

# We need to talk about reliability: making better use of test-retest studies for study design and interpretation

Granville J. Matheson

Department of Clinical Neuroscience, Center for Psychiatry Research, Karolinska Institutet and Stockholm County Council, Stockholm, Sweden

## ABSTRACT

Neuroimaging, in addition to many other fields of clinical research, is both time-consuming and expensive, and recruitable patients can be scarce. These constraints limit the possibility of large-sample experimental designs, and often lead to statistically underpowered studies. This problem is exacerbated by the use of outcome measures whose accuracy is sometimes insufficient to answer the scientific questions posed. Reliability is usually assessed in validation studies using healthy participants, however these results are often not easily applicable to clinical studies examining different populations. I present a new method and tools for using summary statistics from previously published test-retest studies to approximate the reliability of outcomes in new samples. In this way, the feasibility of a new study can be assessed during planning stages, and before collecting any new data. An R package called relfeas also accompanies this article for performing these calculations. In summary, these methods and tools will allow researchers to avoid performing costly studies which are, by virtue of their design, unlikely to yield informative conclusions.

# INTRODUCTION

In the assessment of individual differences, reliability is typically assessed using test-retest reliability, inter-rater reliability or internal consistency. If we consider a series of measurements of a particular steady-state attribute in a set of individuals, the variability between the measured values can be attributed to two components: inter-individual differences in the true underlying value of the attribute, and differences due to measurement error. Conceptually, reliability refers to the fraction of the total variance which is not attributable to measurement error (*Fleiss, 1986*). Hence, it yields information regarding the overall consistency of a measure, the distinguishability of individual measurements, as well as the signal-to-noise ratio in a set of data. Accordingly, a reliability of 1 means that all variability is attributable to true differences and there is no measurement error, while a reliability of 0 means that all variability is accounted for by measurement error. A reliability of 0.5 means that there is equal true and error-related variance (Fig. 1): this implies that, for an individual whose underlying true value is equal to the true group mean,

Corresponding author
Granville J. Matheson,
granville.matheson@ki.se,
mathesong@gmail.com

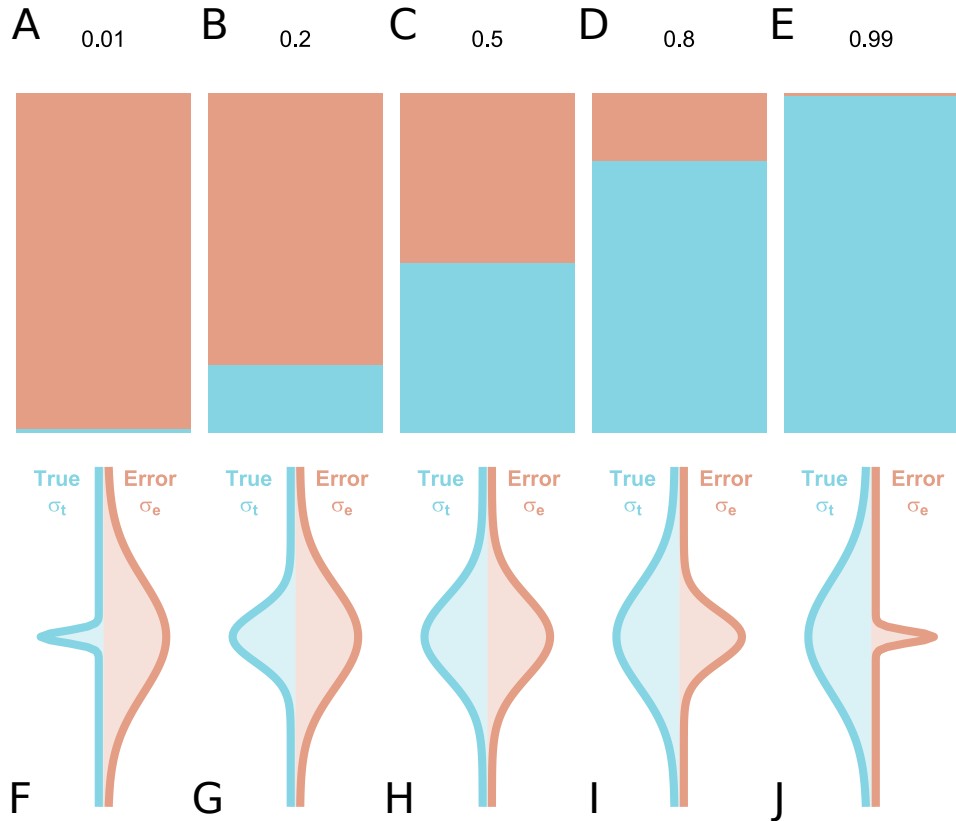

**Figure 1** True inter-individual variance in blue (i.e., between-individual variability of the underlying 'true' values), and measurement error variance in red (i.e., within-individual variability) for different intraclass correlation coefficient (ICC) values, as fractional contributions to the total variance (A–E) and by density distributions showing the size of the distributions relative to one another (F–J).

the long-run distribution of measured values would overlap with the entire population distribution of true values, under ideal conditions. Owing to its relating true and error-related variance, reliability can therefore be increased either by reducing the measurement error, or by increasing the amount of true interindividual variability in the sample such that measurement error is proportionally smaller.

The null hypothesis significance testing (NHST) paradigm for statistical inference is often used in clinical research. According to this approach, a result is considered significant when the *p* value is less than the prespecified alpha threshold, and the null hypothesis is then rejected. In this paradigm, study design is performed by considering the risk of type I and type II errors. In practice, there is a great deal of attention given to the minimisation of type I errors, i.e., false positives. This usually takes the form of correction for multiple comparisons. Another important consideration is the minimisation of type II errors, i.e., false negatives. This takes the form of power analysis: reasoning about the number of participants to include in the study based on the effect size of interest. In contrast, there has been comparatively little consideration given to the reliability of the measures to be used in the study, although there there has been much written on this topic (e.g., *Schmidt & Hunter,*

*1996*; *Kanyongo et al., 2007*; *Loken & Gelman, 2017*). The reliability of outcome measures limits the range of standardised effect sizes which can be expected (although it can also increase their variance in small samples (*Loken & Gelman, 2017*), which is vital information for study design. Assuming constant underlying true scores, outcome measures with lower reliability have diminished power, meaning that more participants are required to reach the same conclusions, that the resulting parameter estimates are less precise (*Peters & Crutzen, 2018*), and that there is an increased risk of type M (magnitude) and type S (sign) errors (*Gelman & Carlin, 2014*). In extreme cases, if measured values are too dissimilar to the underlying 'true' values relative to any actual true differences between individuals, then a statistical test will have little to no possibility to infer meaningful outcomes: this has been analogised as "Gelman's kangaroo" (*Gelman, 2015*; *Wagenmakers & Gronau, 2017*).

Assessment of reliability is critical both for study design and interpretation. However it is also dependent on characteristics of the sample: we can easily use a bathroom scale to contrast the weight of a feather and a brick. In all cases, the scale will correctly indicate that the brick weighs more than the feather. However, we cannot conclude from these results that this bathroom scale can reliably measure the weight of bricks, and proceed to use it to examine the subtle drift in weight between individual bricks produced by a brick factory. In this way, the reliability of a measure is calibrated to the inter-individual differences in that sample. In psychometrics, reliability is often assessed using internal consistency (*Ferketich, 1990*). This involves examining the similarity of the responses between individual items of the scale, compared to the total variability in scores within the sample. This means that the reliability of the scale can be estimated using only the data from a single completion of the scale by each participant. However, for most clinical/physiological measures, estimation of reliability by internal consistency is not possible, as the measurement itself cannot be broken down into smaller representative parts. For these measures, reliability can only be assessed using test-retest studies. This means that measurements are made twice on a set of individuals, and the inter- and intra-individual variability are compared to determine the reliability. If the variability of the outcome is similar in the test-retest and applied studies, it can reasonably be assumed that the measure will be equally reliable in both studies.

When examining clinical patient samples however, it often cannot be assumed that these samples are similar to that of the test-retest study. One solution is to perform a new test–retest study in a representative sample. However, for outcome measures which are invasive or costly, it is usually not feasible to perform test-retest studies using every clinical sample which might later be examined. Positron emission tomography (PET), which allows imaging of in-vivo protein concentrations or metabolism, is both invasive and costly; participants are injected with harmful radioactivity, and a single measurement can cost upwards of USD 10,000. In PET imaging, it is usually only young, healthy men who are recruited for test-retest studies. These samples can be expected to exhibit low measurement error, but may also be expected to show limited inter-individual variability. Despite reporting of reliability in test-retest studies being common practice, when the reported reliability is low, little consideration is often given to these reliability estimates on the basis of insufficient inter-individual variability, i.e., it is assumed that there will be more variation in clinical comparison studies and that the reliability does not accurately

reflect this. This is certainly true in some circumstances. However, when it is not true, it can lead to the design of problematic studies whose ability to yield biologically meaningful conclusions is greatly limited. This is costly both in time and resources for researchers, and leads to the needless exposure of participants to radiation in PET research. It is therefore important to approximate the reliability of a measure for the sample of interest before data collection begins for studies investigating individual differences.

In this paper, I present how reliability can be used for study design, and introduce a new method for roughly approximating the reliability of an outcome measure for new samples based on the results of previous test-retest studies. This method uses only the reliability and summary statistics from previous test-retest studies, and does not require access to the raw data. Further, this method allows for calculation of the characteristics of a sample which would be required for the measure to reach sufficient levels of reliability. This will aid in study planning and in assessment of the feasibility of new study designs, and importantly can be performed *before* the collection of any new data. I will demonstrate how these methods can be utilised by using five examples based on published PET test-retest studies. This paper is also accompanied by an R package called *relfeas* (http://www.github.com/mathesong/relfeas), with which all the calculations presented can easily be applied.

## METHODS

### Reliability

From a classical test theory perspective, observed values are equal to an underlying true value plus error. True scores can never be directly observed, but only estimated. Within this paradigm, reliability relates the degree of variance attributable to true differences and to error.

$$\rho = \frac{\sigma_t^2}{\sigma_t^2 + \sigma_e^2} = \frac{\sigma_t^2}{\sigma_{tot}^2} \tag{1}$$

where $\rho$ denotes reliability, and $\sigma^2$ represents the variance due to different sources (t: true, e: error, and tot: total). This definition of reliability is used both for measures of internal consistency (for which Cronbach's $\alpha$ is a lower-bound estimate), and of test-retest reliability (which can be estimated using the intraclass correlation coefficient, ICC).

Reliability can therefore be considered a measure of the distinguishability of measurements (*Carrasco et al., 2014*). For example, if the uncertainty around each measurement is large, but inter-individual variance is much larger, scores can still be meaningfully compared between different individuals. Similarly, even if a measure is extremely accurate, it is still incapable of meaningfully distinguishing between individuals who all possess almost identical scores.

### *Test–retest reliability*

Test-retest reliability is typically estimated using the ICC. There exist several different forms of the ICC for different use cases (*Shrout & Fleiss, 1979*; *McGraw & Wong, 1996*), for which the two-way mixed effects, absolute agreement, single rater/measurement (the ICC(A,1)

according to the definitions by *McGraw & Wong (1996)*, and lacking a specific notation according to the definitions by *Shrout & Fleiss (1979)* is most appropriate for test-retest studies (*Koo & Li, 2016*).

$$ICC = \frac{MS_R - MS_E}{MS_R + (k-1)MS_E + \frac{k}{n}(MS_C - MS_E)} \qquad (2)$$

where MS refers to the mean sum of squares: $MS_R$ for rows (also sometimes referred to as $MS_B$ for between subjects), $MS_E$ for error and $MS_C$ for columns; and where k refers to the number of raters or observations per subject, and n refers to the number of subjects.

While many test-retest studies, at least in PET imaging, have traditionally been conducted using the one-way random effects model (the ICC(1,1)) according to the definitions by *Shrout & Fleiss (1979)*, the estimates of these two models tend to be similar to one another in practice, especially relative to their confidence intervals. As such, this does not nullify the conclusions of previous studies; rather their outcomes can be interpreted retrospectively as approximately equal to the correct metric.

Importantly, the ICC is an approximation of the true population reliability: while true reliability can never be negative (Eq. 1), one can obtain negative ICC values, in which case the reliability can be regarded as zero (*Bartko, 1976*).

### Measurement error

Each measurement is made with an associated error, which can be described by its standard error ($\sigma_e$). It can be estimated (denoted by a hat ∧) as the square root of the within subject mean sum of squares ($MS_W$), which is used in the calculation of the ICC above (*Baumgartner et al., 2018*).

$$\hat{\sigma}_e^2 = MS_W = \frac{1}{n(k-1)} \sum_{i=1}^{n} \sum_{j=1}^{k} (y_{ij} - \bar{y}_i)^2 \qquad (3)$$

where *n* represents the number of participants, *i* represents the subject number, *j* represents the measurement number, *k* represents the number of measurements per subject, *y* represents the outcome and $\bar{y}_i$ represents the mean outcome for that subject.

The standard error can also be estimated indirectly by rearranging Eq. (1), using the ICC as an estimate of reliability and using the squared sample standard deviation ($s^2$) as an estimate of the total population variance ($\sigma_{tot}^2$). This is often referred to as the standard error of measurement (SEM) (*Weir, 2005*; *Harvill, 1991*).

$$SEM = s\sqrt{1 - ICC} \approx \sigma_e = \sigma_{tot}\sqrt{1 - \rho} \qquad (4)$$

in which s refers to the standard deviation of all measurements in the sample (both test and retest measurements for test-retest studies).

The standard error can be expressed either in the units of measurement, or relative to some property of the sample. It can be: (i) scaled to the variance of the sample as an estimate of the reliability (ICC, Cronbach's $\alpha$), (ii) scaled to the mean of the sample as an estimate of the relative uncertainty (the within-subject coefficient of variation, WSCV) (*Baumgartner et al., 2018*), or (iii) unscaled as an estimate of the absolute uncertainty ($\hat{\sigma}_e$ or SEM).

The relative or absolute uncertainty can be used to calculate the smallest detectable difference (SDD) (also sometimes referred to as the minimum detectable difference) between two measurements in a given subject which could be considered sufficiently large that it is unlikely to have been due to chance alone (say, according to a 95% confidence interval, i.e., using $z_{(1-\alpha/2)} = 1.96$ below) (*Weir, 2005*; *Baumgartner et al., 2018*). This is calculated using the following equation.

$$SDD = \sqrt{2} \times z_{(1-\alpha/2)} \times \hat{\sigma}_e. \tag{5}$$

It can also be extended to a group level examining differences in means.

$$SDD = \sqrt{2} \times z_{(1-\alpha/2)} \times \frac{\hat{\sigma}_e}{\sqrt{n}}. \tag{6}$$

It is important to mention that this measure assumes that all measurements belonging to the original test-retest study and later application study exhibit exactly the same underlying standard error. Further, this measure is not a power analysis: the SDD simply infers that a change in outcome or mean outcome of a group of measurements before and after an experimental manipulation is larger than would be expected by chance. For performing a power analysis for the within-subject change for application to a new study, one must consider the expected effect size relative to the standard deviation of the within-individual changes in the outcome measure.

### Standard thresholds

In psychometrics, reliability can be calculated with one scale by examining its internal consistency. Despite being calculated from the consistency of responding between items, reliability assessed by internal consistency amounts to the same fundamental definition of reliability, namely the ratio of the true to the total variance in the data. It is considered good practice in psychometric studies to calculate the reliability of a scale using measures of internal consistency in the study sample prior to performing statistical inference. The reliability can be used to confirm that the data is sufficiently capable of distinguishing between individuals, for which *Nunnally (1978)* recommended 0.7 as a default lowest acceptable standard of reliability for scales used in basic research, and 0.8 as adequate. For applied settings in which important decisions are made based on measured outcomes, he suggests a reliability of 0.9 as a minimum and 0.95 as adequate, since even with a reliability of 0.9, the standard error of measurement is almost a third the size of the standard deviation.

Test-retest reliability has traditionally been defined by more lenient standards. *Fleiss (1986)* defined ICC values between 0.4 and 0.75 as good, and above 0.75 as excellent. *Cicchetti (1994)* defined 0.4 to 0.59 as fair, 0.60 to 0.74 as good, and above 0.75 as excellent. These standards, however, were defined considering the test-retest (as opposed to internal consistency) reliability of psychometric questionnaires (*Shrout & Fleiss, 1979*; *Cicchetti, 1994*) for which changes from test to retest could be caused by measurement error, but could also be caused by actual changes in the underlying 'true' value. In PET measurement where protein concentrations are estimated, or indeed in many other physiological measures, these within-subject fluctuations can usually be assumed to be negligibly small over short time periods. For this reason, these standards cannot be considered to be directly applicable.

More conservative standards have been proposed by *Portney & Watkins (2015)*, who define values between 0.5 and 0.75 as "poor to moderate", 0.75 to 0.9 as "good", and above 0.9 as acceptable for "clinical measures". These standards correspond more closely with those defined for the internal consistency of psychometric instruments, and can be considered to be more applicable for measures for which 'true' changes are thought to be negligible.

### Relation to effect size estimates

It is important to consider that reliability is related to effect sizes, and more specifically effect size attenuation. This applies both to studies examining correlations with a continuous variable, and those making group comparisons. This relation is described by the following equation (*Spearman, 1904*; *Nunnally, 1970*).

$$r_{ObsA,ObsB} = r_{A,B} \times \sqrt{\rho_A \times \rho_B} \tag{7}$$

where $r_{A,B}$ is the true association between variables A and B, and $r_{ObsA,ObsB}$ is the observed correlation between the measured values for A and B after taking into account their reliability ($\rho$). When comparing two or more measures, each of which are recorded with some degree of imprecision, the combined reliability is equal to the square root of their product. This defines the degree to which an association is attenuated by the imprecision of the recorded outcomes, and sets an upper bound on the strength of association between variables that can be expected (*Vul et al., 2009*). This attenuation is important when performing power analyses as it decreases the range of potential standardised effect sizes which can be expected.

Although smaller standardised effect sizes are to be expected with increasing measurement error, such decreases are only to be expected on average. *Loken & Gelman (2017)* point out that greater measurement error also leads to greater variation in the measured effect sizes which, particularly for smaller sample sizes, also means that some measured effect sizes overestimate the true effect size by chance. This is exacerbated by publication bias, i.e., the selection of tested effects or hypotheses which are found to be statistically significant. This can even lead to a higher proportion of studies being reported whose estimates exaggerate the true underlying effect size, despite the attenuation on average as a result of poor reliability, if only those results which reach statistical significance are selected.

When planning a study, sample size is often determined using power analysis: this means calculating the required sample size to detect a given effect size a given proportion of the time. The effect size is sometimes specified as equal to that reported by previous studies; however, this makes studies vulnerable to the accuracy by which the effect size is approximated (*Simonsohn, 2015*; *Morey & Lakens, 2016*). A better approach is to power studies for the smallest effect size of interest such that all interesting effects are likely to be detected. More conservative still is the suggestion to power studies according to the level at which meaningful differences between individual studies can be reliably detected (*Morey & Lakens, 2016*). These strategies would ensure that the scientific literature would produce robust findings, and that the effects in individual studies can be statistically compared, respectively. However, for costly and/or invasive methods, such ideals are not

usually attainable, or ethical. Rather, for studies with these constraints, we require a lower standard of evidence, with strong conclusions perhaps left for meta-analyses (or better yet, for meta-analyses of pre-registered studies to avoid effect size inflation for statistically significant results due to low power (*Cremers, Wager & Yarkoni, 2017*) or publication bias (*Ferguson & Heene, 2012*) and not drawn from statistically significant findings from individual studies alone. To determine the feasibility of an individual study using these methods, it may be more useful not to consider the point at which an individual study is *ideal* for the scientific literature as described by *Morey & Lakens (2016)*, but rather the point at which the study will simply *not be bad*, i.e., will at least be capable of answering the research question at hand given realistic assumptions.

For this latter aim, we can perform power analysis for study feasibility assessment in terms of a maximum realistic strength of association in biological terms. In this way, the corresponding effect size can be calculated after taking the effect size attenuation into account. If a study will be insufficiently powered even for such a strength of association, then the study is certainly miscalibrated for the detection of the effect of interest. In this way, this method is useful for deciding which research questions can be rejected out of hand given the relevant constraints on the sample size.

## Extending test-retest reliability

In order for the results of previous test-retest studies to be applicable for new samples with different characteristics, we need to be able to extrapolate the reliability for new samples. To this end, I propose the use of the extrapolated ICC (below). This calculation is based upon two assumptions.

### Assumption 1

The absolute measurement error ($\sigma_e$) will be either similar between groups and/or studies, or the extent to which it will be larger or smaller in a new sample can be approximated.

*Comment.* Since test-retest studies are typically conducted using young, healthy volunteers, and not patient samples, these groups will usually provide a good estimate at least of the maximum bound of the precision of the measurement. For comparison groups for which measurement error is likely to be larger, it can either be assumed to be similar (a liberal assumption), or the extent to which it is larger can be approximated as a small multiple (a conservative assumption, e.g., 20% larger in the patient group). While an accurate estimation of this calibration factor can be difficult in practice, a worst-case-scenario estimation can be helpful as in *Example 1* and Appendix 2.

### Assumption 2

The absolute measurement error (SEM) is likely to be relatively stable across different ranges of the same outcome measure (i.e., an individual with a low score will have the same SEM as an individual with a high score on the same outcome measure).

*Comment.* There is often some variation expected in the measurement error for different values of the outcome (e.g., more error for individuals with a higher value of the outcome, or vice versa); however, this variation is usually expected to be fairly small. If, on the other

hand, this variation across the range of the outcome measure *is* expected to be substantial, then additional validation studies are likely warranted to determine that such values of the outcome measure can be estimated with a sufficient degree of precision.

### Assumption 3
The ICC, and its extension here, make the assumption of normality of the data.

*Comment.* While there are methods available for the estimation of the reliability of non-Gaussian data (*Nakagawa & Schielzeth, 2010*), this method, as defined, is not capable of extrapolating reliability in such cases.

### Assumption summary
These assumptions are reasonable given that the goal is a rough approximation of the reliability in a new sample in order to assess feasibility in study design and interpretation, under the assumption of normality. The first two assumptions are primarily based on the fact that most previous and current test-retest studies are performed on young, healthy participants. For planning a new study for which large deviations are expected from either of these two assumptions, one approach could be to make approximations based on best- and worst-case scenarios. If such an approach is not possible for whichever reason, this speaks in favour of simply conducting a new test-retest study to directly evaluate the feasibility of such a study design in the specific context of the proposed study.

### The extrapolated ICC
The ICC can be expressed using the SEM and the SD (s) of the original study from Eq. 4.

$$ICC = 1 - \frac{SEM^2}{s^2}. \tag{8}$$

Given these two assumptions above, we can approximate the ICC for a new sample given only the standard deviation of the new sample (and an approximation of the inflation extent of the SEM):

$$ICC_{NewStudy} = 1 - \frac{(\tau SEM_{TRT})^2}{s^2_{NewStudy}} = 1 - \frac{\tau^2 s^2_{TRT}(1 - ICC_{TRT})}{s^2_{NewStudy}} \tag{9}$$

where $\tau$ represents the error (SEM) inflation factor, representing the multiplicative increase in expected measurement error in the new study, for whichever reason. Using $\tau = 1$ assumes that the SEM is the same between studies, while $\tau = 1.2$ would assume a 20% increase in the measurement error. If $\tau$ is thought to be substantially greater than 1, then a new test-retest study is perhaps warranted. With $\tau = 1$ or $\tau \approx 1$, the difference between the ICC of the original study and the extrapolated ICC of a new study will be determined primarily by the SD of the new study.

### Studies making use of continuous independent variables (correlations)
For a new study which is going to examine a correlation, reliability is simply determined by the variance of the study group. With more variation in the study sample, the reliability is therefore higher, all else being equal. Increasing sample variability can be attained by implementing wider recruiting strategies, and not simply relying on convenience sampling
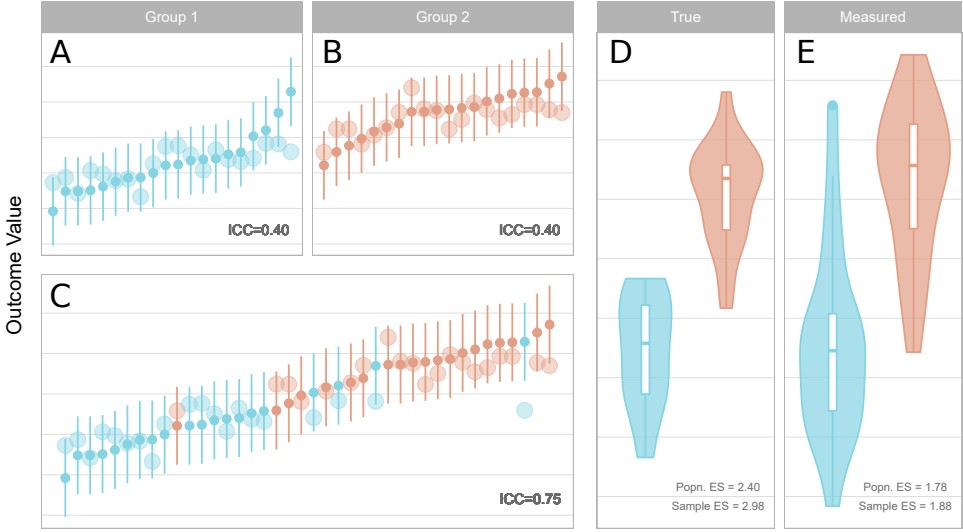

**Figure 2** **Reliability, measurement error, inter-individual heterogeneity, group differences and effect size attenuation as they relate to one another.** (A–C) Measured values and their 95% confidence intervals are represented by the small points and error bars, and underlying true values are represented by the larger points. When comparing groups with a sufficiently large effect size, low reliability within groups (A, B) is increased for the total sample (C) due to the larger variance of the combined sample. (D–E) Standardised effect sizes (ES, Cohen's d) between groups for true underlying values (D, i.e. without measurement error) are attenuated by measurement error in the measured values (E). The population (Popn.) ES refers to the underlying distributions from which the data are sampled, and sample ES refers to the obtained sample represented in A–C. (A) Measured values and their 95% confidence intervals are represented by the small points and error bars, and underlying true values are represented by the larger points. When comparing groups with a sufficiently large effect size, low reliability within groups (left, upper panels) is increased for the total sample (left, lower panel) due to the larger variance of the combined sample. (B) Standardised effect sizes (ES, Cohen's d) between groups for true underlying values (left, i.e., without measurement error) are attenuated by measurement error in the measured values (right). The population (Popn.) ES refers to the underlying distributions from which the data are sampled, and sample ES refers to the obtained sample represented in the leftmost panels.

(see *Henrich, Heine & Norenzayan, 2010*). This also has the advantage of increasing the external validity of findings.

It should be noted, however, that increasing sample variability by including individuals who differ on another variable which is known to be associated with the dependent variable (for example, age) would *not* necessarily increase the reliability despite increasing the variance of the study group. In these cases, this variable should be included in the statistical test of an applied study as a covariate, and hence the variance of the unstandardised residuals *after* accounting for the covariate would therefore be a better estimation of the total variance for reliability analysis in this test-retest sample.

### Studies making use of binary independent variables (t-tests)

For a new study which is comparing two independent groups, one is fitting a binary regressor to the dependent outcome variable. The SD for the ICC is calculated based on the sample variability before fitting the regressor, and thus all individuals are included in the calculation of this value (Fig. 2). The total SD is therefore dependent on both the

within-group standard deviation, as well as the degree of difference between the two groups, which is measured by the effect size. For independent sample t-tests, one can therefore calculate the effect size (Cohen's d) for which the reliability of a measure would reach a certain desired threshold.

Thus, by estimating the within-group standard deviation of each of the two groups, one can calculate the required effect size to obtain a sufficient level of reliability. The total SD of the entire sample (including both groups) is described by the following equation:

$$\sigma_{total} = \sqrt{\frac{(n_1-1)\sigma_1^2 + n_1\mu_1^2 + (n_2-1)\sigma_2^2 + n_2\mu_2^2 - (n_1+n_2)\mu_{total}^2}{n_1 + n_2 - 1}} \tag{10}$$

where $n$ is the number of participants in each group, $\mu$ is the mean of each group, $\sigma$ is the SD of each group, and the subscripts 1, 2 and total refer to group 1, group 2, and the combined sample. The total mean ($\mu_{total}$) is calculated as follows.

$$\mu_{total} = \frac{1}{n_{total}}(n_1\mu_1 + n_2\mu_2). \tag{11}$$

We therefore only need to solve this equation for $\mu_2$, the mean of the second (e.g., patient) group. This can be calculated as follows (separating the equation into three parts):

$$A = \mu_1^2 n_1 n_2 + n_1^2\sigma_1^2 - n_1^2\sigma_{total}^2 + n_1 n_2\sigma_1^2 + n_1 n_2\sigma_2^2 - 2n_1 n_2\sigma_{total}^2 - n_1\sigma_1^2 \tag{12}$$

$$B = -n_1\sigma_2^2 + n_1\sigma_{total}^2 + n_2^2\sigma_2^2 - n_2^2\sigma_{total}^2 - n_2\sigma_1^2 - n_2\sigma_2^2 + n_2\sigma_{total}^2 \tag{13}$$

$$\mu_2 = \frac{1}{2n_1 n_2} \times \left(2\mu_1 n_1 n_2 \pm \sqrt{4\mu_1^2 n_1^2 n_2^2 - 4n_1 n_2(A+B)}\right). \tag{14}$$

In this way, given the group sizes and estimates of the standard deviation within both groups, we can calculate the mean difference and effect size for which the degree of variability in the total sample is sufficient to reach a specified level of reliability.

### Software
All analyses in this paper were performed using R (*R Core Team, 2018*, *Joy in Playing*) with the *relfeas* package (https://github.com/mathesong/relfeas, v0.0.2).

## RESULTS
The use of these methods will be demonstrated using examples based on two PET tracers for which test-retest studies have been published. PET imaging involves injection of a radioligand (or tracer) into the blood which binds to a target of interest, and the concentration of radioactivity in the tissue, and sometimes also the blood, are measured over time. The two tracers used as examples include [$^{11}$C]AZ10419369 for imaging of the serotonin 1B receptor (5-HT$_{1B}$) and [$^{11}$C]PBR28 for imaging of translocator protein (TSPO). Both of these tracers have been well-validated and have been used to make clinical

comparisons. The reason for the selection of these two radioligands is that they both demonstrate properties which make the interpretation of reliability outcomes especially difficult. [$^{11}$C]AZ10419369 exhibits low measurement error, but also low inter-individual heterogeneity—even lower than the measurement error—resulting in low reliability. [$^{11}$C]PBR28 shows high error variance, but also high inter-individual heterogeneity—over three times greater than the measurement error—resulting in high reliability.

Outcome measures reported for these PET studies include binding potential (BP$_{ND}$), volume of distribution (V$_T$), which are defined based on compartmental models, as well as the distribution volume ratio (DVR) and standardised uptake value (SUV). For more details, see Appendix 1.

### Example 1: reliability can be approximated for applied studies from a test-retest validation study with lower inter-individual variance

[$^{11}$C]AZ10419369 is a radiotracer for the serotonin 1B (5-HT$_{1B}$) receptor, which is widely expressed in the brain and an important target in depression (*Ruf & Bhagwagar, 2009*; *Tiger et al., 2018*). In the test-retest study published with this ligand (*Nord et al., 2014a*; *Nord et al., 2014b*), using the frontal cortex as an example, it showed what is considered a high mean BP$_{ND}$ (1.6), a favourable absolute variability (6.8%) and a good coefficient of variation (CV, 6–7%). However in the sample measured, the ICC was very low: 0.32. It was concluded that "[the low ICC value] can be explained by…a low between-subject variance. Thus, despite the low ICC values, it cannot be excluded that the test-retest reliability is also high in these regions" (p. 304).

A followup study was published examining age-related changes in 5-HT$_{1B}$ binding (*Nord et al., 2014a*; *Nord et al., 2014b*), concluding that 5-HT$_{1B}$ receptor availability decreases with age in cortical regions. If one were to have considered the ICC of the original study as a fixed property of this measure, then such a conclusion should be treated with some degree of caution. The low reliability of the outcome means that the observed strength of association will, on average, be reduced compared to the true association (thereby reducing power); but the observed relationship will also show a larger variation and could very well overestimate the true association (*Loken & Gelman, 2017*). However, due to the differences in [$^{11}$C]AZ10419369 BP$_{ND}$ across the age range examined, the standard deviation was 3.2-fold greater than that of the test-retest study. To assess the reliability of frontal cortex [$^{11}$C]AZ10419369 BP$_{ND}$ for this particular application, it is necessary to take the greater variability into account.

We can calculate the reliability of the new study using the results of the test-retest study by assuming the same measurement error between the studies. In this way we obtain a reliability of this tracer and for this particular correlation equal to 0.93. From this analysis, the outcome of the study can be considered reliable in terms of measurement error, and that individuals can be easily distinguished from one another by their outcome measures. This conclusion also holds after taking into account the use of partial volume effect correction in this study (see Fig. 3 and Appendix 2).
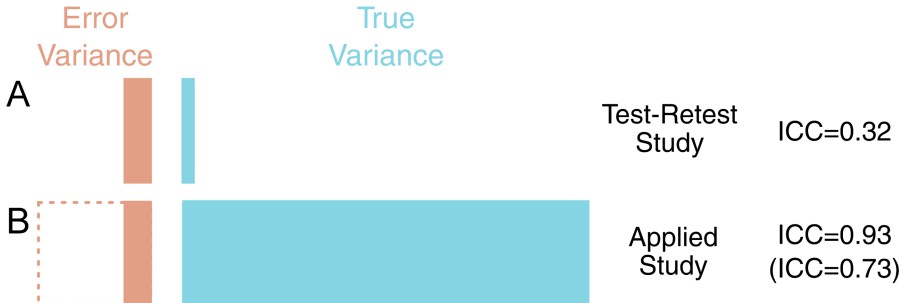

**Figure 3** **Demonstration of the outcomes of Example 1.** Left are shown the proportional contributions of error and true variance to the total variance for the different studies (Test Retest A, Applied B). The difference in the variance of different samples can be extreme, and the results of test-retest studies examining young, healthy individuals can require re-calibration when examining new samples with very different characteristics, in this case spanning a wide age range. Below left, shown with dashed lines, is shown that even in a worst-case scenario of measurement error ($\hat{\sigma}_e$) being doubled in this new sample (and thereby variance quadrupled), this can still be compensated for by the larger true variance (ICC right in brackets).

## Example 2: accounting for reliability is important when performing power analysis

Let us assume that a new study was being planned to examine the relationship between [11C]AZ10419369 BP$_{ND}$ and a particular self-report measure. This study would be performed in young, healthy control subjects. Having observed from the test-retest study that the reliability in healthy controls was not high for most regions due to limited inter-individual variability, the investigators choose to examine the occipital cortex where reliability is highest (ICC = 0.8).

The self-report scale is also associated with its own measurement error: it usually exhibits a reliability of 0.7 in healthy samples (and it is assumed that this will apply to this sample too). Using Eq. 7, we obtain an attenuation of the correlation coefficient of 0.75. In other words, if the scale were to explain 100% of the variance in [11C]AZ10419369 BP$_{ND}$ in the absence of any measurement error, we would expect to see a correlation of r=0.75 after taking this error into account.

For deciding whether to go ahead and conduct this study, we can consider the *maximum realistic biological* effect size for power analysis (see 'Methods'). The investigators approximate that 30% explained variance is probably the maximum extent to which the self-report measure could realistically be explained by occipital cortex 5-HT$_{1B}$ availability. For this effect, a power analysis (for 80% power and a significance level of 0.05) will reveal that such a correlation would require a sample size of 24 in the absence of measurement error. Accounting for the attenuation due to measurement error however, 44 participants would be required to observe this effect.

It can therefore be concluded that unless more than 44 participants can be examined in the planned study, then it will not be possible to draw strong conclusions from the results of this study. If significant, it would likely overestimate the true magnitude of the association as the study would only be sufficiently powered for effect sizes larger than what was considered realistic a priori, i.e., a type M error (*Gelman & Carlin, 2014*). If

non-significant, such a result would be mostly uninformative as the type II error rate would be too large, although its (wide) confidence intervals might help to inform a later meta-analysis. In this way, a costly study which had a low probability of yielding meaningful conclusions can be avoided before even its inception.

## Example 3: robust within-individual effects do not imply high reliability for between-individual comparisons

$[^{11}C]AZ10419369$ is also known to be displaced by serotonin release. This can be measured by performing a baseline measurement, and then performing another measurement following administration of a serotonin-releasing agent and assessing the decrease in binding (see Appendix 3). We will assume that high doses of the drug similar to those previously administered to non-human primates (NHPs) (*Nord et al., 2013*) will be administered to humans, and it is estimated that this will produce a similar displacement of the radioligand in humans. The study will consider the occipital cortex, where the reliability of $[^{11}C]AZ10419369$ $BP_{ND}$ is highest, and where escitalopram-induced displacement was significant in NHPs (mean $= -12\%$, SD $= 10\%$) (*Nord et al., 2013*).

For this study, there are two questions to be answered with regard to its feasibility, namely the within- and between-individual effects respectively. The first is whether we can reliably measure serotonin release within individuals at all. The second question is whether the reliability of this radioligand is sufficiently high to compare the degree of radioligand displacement (i.e., $\Delta BP_{ND}$), as opposed to binding (i.e., $BP_{ND}$), between individuals and hence groups (patients vs controls).

The first question can be answered using the smallest detectable difference (SDD) metric as well as power analysis. The SDD for individual scores is 14.3% (Eq. (5)). To detect a difference in group means, the SDD is 10.1% for 2 individuals (Eq. (6)). This means that with two or more individuals, a change of 12% is greater than that which would be expected by chance based on the reliability of $[^{11}C]AZ10419369$ $BP_{ND}$. We also perform a power analysis by considering the size of the expected effect ($-12\%$ change) relative to the standard deviation of test-retest variability (7.3%), i.e., a Cohen's d of 1.64. This means that 5 individuals would be required for 80% power, and a 0.05 significance level, for a one-sided test of this effect size, assuming that it is accurate.

The second question requires that we reassess the reliability of the radioligand for the change in $[^{11}C]AZ10419369$ $BP_{ND}$. We must consider that there are two measurements, both of which have error terms: the error ($\sigma_e$) term is therefore doubled for $\Delta BP_{ND}$ as the measurement error is additive. The total variance term is also no longer determined by the range of $BP_{ND}$ values, but by the range of $\Delta BP_{ND}$ values. This can be approximated from the displacement statistics as 10% of the mean $BP_{ND}$ value before treatment. The ICC obtained for $\Delta BP_{ND}$ would therefore be $-0.06$. This is so low due to the low variability in the outcome: there is large error variance, and a small true variance in the $\Delta BP_{ND}$ outcome measure. However this does not take the expected differences between the controls and the patients into consideration.

In planning this study, the investigators estimate that there should be 2.5 times as much serotonin release in the patient group. This will result in a larger degree of interindividual

differences, and thus higher reliability. With two groups of equal size (20) and standard deviation, with this effect size, the ICC grows to 0.41. This means that the true effect (Cohen's d=1.8) will be attenuated to 0.95 when measured due to the measurement error, and a much larger sample size (308% for unpaired; 228% for paired) will be required to detect a difference of this size as significant with 80% power and 0.05 significance level (with a one-sided test). Further, since differences between individuals in $\Delta BP_{ND}$ values will be mostly determined by random noise, even if the study were to produce a significant outcome, it would still be unclear to what extent this was due to noise, and we would remain uncertain about its true magnitude. It can therefore be concluded that this study is probably not worth investing in, as its results will not be greatly informative.

### Example 4: High reliability does not imply sensitivity for small proportional within-individual differences

[$^{11}$C]PBR28 is a second generation tracer for translocator protein (TSPO), which is expressed in glial cells including immune cells, and represents an important target for studies of immune activation in various clinical disorders. Due to genetic effects (leading to high- and low-affinity subtypes of the protein) and high intra- and inter-individual variability, several different strategies have been proposed for quantification of this tracer.

While the absolute variance of [$^{11}$C]PBR28 $V_T$ appears very high ($\approx 20\%$), its test-retest reliability is also very high ($\approx 0.9$) (*Collste et al., 2016*; *Matheson et al., 2017*) in both high-affinity binders (HABs) and medium affinity binders (MABs) (for more information, see *Owen et al., 2012*) due to the several-fold variability (CV $\approx 40\%$) between individuals (*Collste et al., 2016*; *Matheson et al., 2017*). As such, clinical studies comparing patients and controls using $V_T$ are unlikely to suffer from poor reliability. However, a large effect size (Cohen's d $= 0.8$) corresponds to a mean difference of over 30% in [$^{11}$C]PBR28 $V_T$, while an equivalent large effect size for [$^{11}$C]AZ10419369 frontal cortex $BP_{ND}$ is less than 6%. Put differently, to detect a 10% difference in binding between groups, for 80% power and a significance level of 0.05, 278 (HAB) participants would be required per group for [$^{11}$C]PBR28, compared to 7 per group for [$^{11}$C]AZ10419369. Large numbers of participants will be required to observe even relatively large proportional differences in [$^{11}$C]PBR28 binding. However, large proportional differences are also more likely for [$^{11}$C]PBR28 due to the large differences between individuals.

### Example 5: variance reduction strategies must be validated to ensure that reliability remains sufficient for inter-individual comparisons

Attempts have been made to account for this variability in [$^{11}$C]PBR28 $V_T$, both within and between genotype groups, with suggestions of simplified ratio-based approaches such as the distribution volume ratio (DVR) despite the absence of a reference region (*Lyoo et al., 2015*). These methods were also attractive as they functionally correct for the plasma free fraction ($f_P$) (see Appendix 4). We previously reported high interregional associations (R $\geq$ 0.98), including denominator regions, resulting in poor reliability for this outcome measure (mean ICC $= 0.5$) (*Matheson et al., 2017*). While we advised caution in the interpretation of results making use of ratio methods for inter-individual effects, it is still theoretically possible that ratio methods might show good reliability for the detection of very large

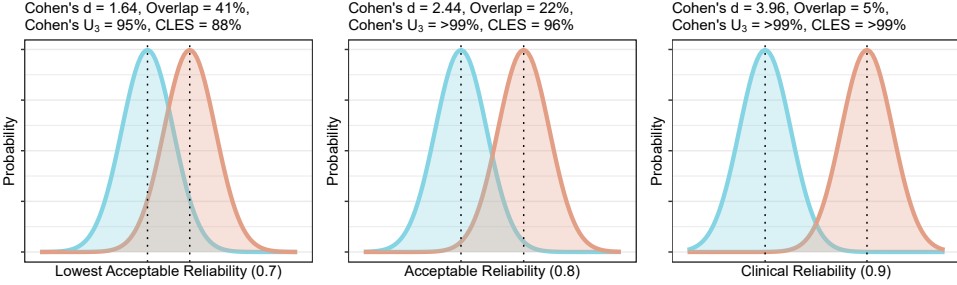

**Figure 4 Overlap and effect size required for DVR differences between groups to reach different levels of reliability (lowest acceptable A, acceptable B, and clinical C).** The overlap is the distributional overlap. Cohen's $U_3$ is the fraction of one group which does not overlap with the mean of the other. The Common Language Effect Size (CLES) is the chance that a person picked at random from the red group will have a higher value than a person picked at random from the blue group.

effects such as in specific neurological conditions which would not be otherwise detectable with small samples due to the large inter-individual variability (*Lyoo et al., 2015*). Using this method of extrapolating reliability, we can calculate what size of an effect between groups would be required for the ratio methods to begin to show reasonable reliability for informative conclusions.

For this, we first calculate the required total SD for each of our specified levels of reliability. We then calculate the effect size which would produce this SD. The methods are calculated for a planned comparison of 20 healthy controls and 20 patients with equal standard deviation between groups, assuming that the SD of the test-retest sample is representative of that for the healthy group. For reliabilities of 0.7, 0.8 and 0.9, one would require Cohen's d effect sizes of 1.6, 2.4 and 4 respectively. *Cohen (1988)* recommended that, in the absence of a known effect size distribution, that a Cohen's d of 0.8, representing a group overlap of 67%, for which 79% of the group with the higher mean would lie above the mean of the group with the lower mean (Cohen's U3), could be considered a *large* effect size. Very similar values for small, medium and large effect sizes were shown for physiological measures (*Quintana, 2017*), and such definitions appear to be well-suited for the effect sizes of molecular imaging studies in psychiatry too (*Howes et al., 2012*; *Kambeitz et al., 2014*; *Gryglewski et al., 2014*). As such, these effect sizes, representing Cohen's U3 of >94% in all cases, can be considered to be unrealistically large. These effect sizes can be plotted as overlapping distributions to provide a visual reference for understanding their magnitude, and to gauge their feasibility using domain knowledge (Fig. 4).

From this information, it is clear that unless one is expecting *very* large effects, the use of DVR is unlikely to be useful for between-individual comparisons due to its poor reliability. Due to the strong associations reported between numerator and denominator regions, DVR effect sizes will even further underestimate actual differences in $V_T$ in addition to the attenuation of effect sizes due to its poor reliability. $V_T$ therefore appears to be statistically superior to DVR for detection of any effects, without the need to be concerned about issues of reliability (although see Appendix 4).

## DISCUSSION

This paper has demonstrated how the results of previously published test-retest studies can be used for the rough approximation of the reliability in new samples based on commonly reported summary statistics and appropriate adjustments. Test-retest studies are usually, as is appropriate, performed to validate new PET tracers and quantification strategies. However, while reliability is an important consideration for the design and interpretation of studies, measures of reliability reported in test-retest studies are not currently used to their full potential. This paper outlines a method for how reliability can be extrapolated to new samples, provides examples of how these methods can be applied in practice for both study design and interpretation, and is accompanied by an open-source R package *relfeas* with functions for all calculations.

In this way, researchers can avoid performing costly studies whose results are unlikely to be replicable or sufficiently informative with regard the relevant biological question being asked, and can interpret the estimates of previously published studies whose reliability might be questionable with an appropriate degree of caution (*Loken & Gelman, 2017*). This method is analogous to the philosophy of "fail fast" from software development (*Shore, 2004*): systems should be designed such that bugs which may lead to system failure will cause the system to fail at an early stage of operation, rather than unexpectedly at a late stage. In the same way, this method allows for studies to fail early, before even reaching the data collection stage, leading to savings in cost and time, and with PET, avoiding exposure of participants to radioactivity.

The importance, and neglect, of reliability has been brought up before, perhaps most notably in the study of *Vul et al. (2009)* examining fMRI studies of emotion, personality and social cognition. This study found 'puzzlingly high correlations', in which, given the reliability of the measures, over 100% of the biological variance was being explained. It was concluded that "a disturbingly large, and quite prominent, segment of fMRI research … is using seriously defective research methods … and producing a profusion of numbers that should not be believed." (p. 285). The failing of researchers to sufficiently acknowledge considerations of reliability has been related to the historical separation of correlational (i.e., individual differences research) and experimental (i.e., within-subject) approaches to scientific inquiry: outcome consistency with one approach does not necessarily apply to the other (*Hedge, Powell & Sumner, 2017*). It was shown that several well-validated cognitive tasks (e.g., Stroop, Go/No-go), despite demonstrating robust within-individual effects, exhibited very low reliability for inter-individual comparisons. This was due primarily to low inter-individual variation. It is noted that it is likely this same characteristic that both makes these measures robust in experimental research, but unreliable for correlational research (*Hedge, Powell & Sumner, 2017*). In sum, between- and within-individual consistency of an outcome measure is optimal for within-individual comparisons, while outcome measures exhibiting consistency of within-individual effects but high between-individual variation are best suited for between-individual comparisons.

The results from the examples above can be interpreted in this context: $[^{11}C]AZ10419369$ $BP_{ND}$ appears to be an excellent outcome for the assessment of within-individual

changes, such as in PET blocking studies where the receptor is measured before and after pharmacological blockade. However, its utility is limited for the assessment of between-individual differences. The opposite is true of $[^{11}C]$PBR28 $V_T$; this measure differentiates between individuals very well, but may be less useful for assessing within-individual changes. $[^{11}C]$PBR28 DVR may be more useful for within-individual effects, provided that equivalence is shown in the denominator region (*Lakens, 2017*). For between-subjects designs, it is important to emphasise that measures whose precision is poor can still be useful when differences between individuals are very large (e.g., $[^{11}C]$PBR28 $V_T$), while measures whose precision is excellent can be practically uninformative when the differences between individuals are extremely small (e.g., frontal cortex $[^{11}C]$AZ10419369 $BP_{ND}$). Since reliability is indirectly proportional to the amount of inter-individual variability in a sample, reliability can be increased by recruiting more diverse participant samples. In contrast, a reliance on convenience sampling can result in study samples which are overwhelmingly comprised of university students from privileged backgrounds (WEIRD: Western, Educated, Industrialized, Rich, and Democratic) (*Henrich, Heine & Norenzayan, 2010*), who are unlikely to be representative of the broader population (i.e., showing low external validity).

Failure to replicate is a substantial problem in science (*Begley & Ellis, 2012*; *Open Science Collaboration, 2015*). Questionable research practices (QRPs) (*John, Loewenstein & Prelec, 2012*; *Simmons, Nelson & Simonsohn, 2011*) and underpowered studies (*Button et al., 2013*; *Morey & Lakens, 2016*) have been proposed as some of the primary reasons for this failure. It has even been suggested that this may threaten the very notion of cumulative science (*Morey & Lakens, 2016*; *Elk van et al., 2015*). Consideration of reliability is another important, and related, factor. Studies based on extremely unreliable outcomes are unlikely to yield estimates which are representative of any underlying reality, especially in the presence of publication bias, which threatens even the potential of meta-analysis to resolve discrepancies (*Elk van et al., 2015*; *Inzlicht, Gervais & Berkman, 2015*). Studies can be determined not to be worth undertaking before their initiation if outcomes are too unreliable, or if maximal feasible 'true' effects after effect size attenuation are deemed to be too small to measure. This can provide enormous savings in both time and resources. A utopian science would be one in which (i) analysis plans were fully pre-registered or, better yet, peer-reviewed prior to study initiation such as in Registered Reports (*Nosek, Spies & Motyl, 2012*; *Chambers et al., 2015*), (ii) studies were sufficiently powered to provide meaningful effect size estimates and comparison between studies (*Morey & Lakens, 2016*); and (iii) all studies utilised outcome measures whose reliability was sufficient for the specific research question in the study sample such that they would yield informative estimates of magnitude. By providing methods such that the reliability in study samples can at least be roughly approximated and planned for, and tools by which to make this easily implementable, this paper hopes to aid researchers in this last goal.

## CONCLUSIONS

Accounting for the reliability of outcome measures is critical when designing new studies, and failure to do so can result in data being collected which are incapable of answering

the specified research question. Reliability is related to the degree of true inter-individual variability in a sample, and testing in different (e.g., patient) samples complicates this issue. In this paper, I present a method by which the reliability of an outcome measure can be approximated for different samples using summary statistics from previous test-retest studies, thereby allowing for analysis of the feasibility of performing new studies using published reports without requiring access to the original data. I demonstrate how reliability can, and should, be accounted for when designing studies, and how this can be applied through five case studies from the field of PET, and provide the tools by which these calculations can be performed. Application of these methods and tools will aid researchers both in designing better studies, as well as determining when a research question cannot be satisfactorily answered with the available resources.

## ACKNOWLEDGEMENTS

I would like to say thank you, in alphabetical order, to Simon Cervenka, Lieke de Boer, Vincent Millischer, Pontus Plavén-Sigray, Björn Schiffler, Jonas Svensson and William Hedley Thompson for their helpful comments and discussions about the manuscript.

## APPENDICES

### Appendix-1: outcome measures

$BP_{ND}$ is a measure of the binding potential: a measure of concentration of the radioligand in the specific (S) compartment of the target tissue relative to sum of free (F) and non-specific (NS) compartments (together defined as the nondisplaceable (ND) compartment). When $BP_{ND}$ is zero, there is no S binding; and when it is 1, there is an equal radioligand concentration in the S and ND compartments. $V_T$ is the total distribution volume: a measure of the total uptake in the target tissue (S + NS + F) relative to the concentration of radioligand in blood plasma. It is therefore a measure of the uptake in the target tissue relative to the exposure of that tissue to the radioligand. DVR is the distribution volume ratio, and is the ratio of $V_T$ values from a target (numerator) and reference (denominator) region of the tissue. SUV is the standardised uptake value: a measure of the total uptake in the target tissue relative to the amount of tracer injected and the body mass. This measure is sensitive to differences in tissue exposure, due to peripheral binding for example, which is assumed to scale with body weight. These outcomes range from more to less specific in this order, and different outcomes are used for different circumstances depending on the fulfilment of certain assumptions underlying them, as well as whether arterial plasma radioactivity concentrations were recorded. More information can be found in *Innis et al. (2007)*.

### Appendix-2: Example 1 Caveat

There is one caveat: the correlational study of age was performed using partial volume effect (PVE) correction. This procedure is used to minimise the bias due to differences in brain volumes and usually produces higher $BP_{ND}$ estimates. However it can also introduce more noise to the data (i.e., a higher SEM). Without access to the original test-retest

data, this increase in SEM ($\tau$) would have to be approximated by assuming that the true between-subject variance is similar between the test-retest sample and the subset of participants from the correlational study of similar age to the test-retest sample (i.e., $\tau \approx CV_{NewStudySubset}/CV_{TRT}$). Another possibility would be to consider the reliability for possible values of $\tau$. In this way, it can be shown that even in the extreme case that the SEM were expected to be doubled following PVE correction, the extrapolated reliability would remain sufficiently high at 0.73.

### Appendix-3: example 3 caveat

Radioligand displacement by endogenous or exogenous substances is common in PET studies, resulting in a decrease in binding. However, it is worth noting that clinical doses of SSRIs have been found to produce a paradoxical increase in [$^{11}C$]AZ10419369 $BP_{ND}$ in humans (*Nord et al., 2013*) for reasons beyond the scope of the current paper. Large doses of SSRIs have been found to produce decreases in NHPs though (*Nord et al., 2013*). For convenience, we will assume that high doses similar to those administered to NHPs will be administered, and assume that it will produce the same change in humans.

### Appendix-4: example 5 caveats

It has been argued that [$^{11}C$]PBR28 $V_T$ should be corrected for the plasma free fraction ($f_P$), i.e., $\frac{V_T}{f_P}$ (*Lyoo et al., 2015*). However, $f_P$ tends to be difficult to measure accurately for [$^{11}C$]PBR28, and while it theoretically increases the validity of the measure, it does so at the cost of diminishing its reliability (*Lyoo et al., 2015*; *Park et al., 2015*). While it has not been conclusively shown that this correction is beneficial, it should be noted that the use of $V_T$ is not necessarily a perfect outcome measure, especially considering that $V_{ND}$ may even exhibit a substantial variability between individuals (*Plavén-Sigray et al., 2019*).

It should also be mentioned that *Lyoo et al. (2015)* observed a CV value for DVR in healthy controls which was more than twice that observed in *Matheson et al. (2017)*. This was also true of the standardised uptake value ratio (SUVR): a simpler measure for which arterial plasma concentrations are not even taken into consideration. This suggests that there were important differences between these data sets, perhaps as a result of differences in ages between samples (23.9 ± 3.0 vs 55.1 ± 15.3 years), recruitment strategies, or because of differences in the measurement length (90 min vs 63 min) and SUV time interval (60–90 min vs 40–60 min). While it is not easy to infer from these results how different the measurement error might have been in these different conditions without a new test-retest analysis, these results do suggest that it is possible that the DVR in the sample used in *Lyoo et al. (2015)* had a higher reliability compared to the *Matheson et al. (2017)* sample.

### Funding

The authors received no funding for this work.

## Competing Interests

The authors declare there are no competing interests.

## Author Contributions

- Granville J. Matheson conceived and designed the experiments, analyzed the data, contributed reagents/materials/analysis tools, prepared figures and/or tables, authored or reviewed drafts of the paper, approved the final draft.

## Data Availability

Data is available at GitHub (v0.0.2: https://github.com/mathesong/relfeas).

## Supplemental Information

Supplemental information for this article can be found online at http://dx.doi.org/10.7717/peerj.6918#supplemental-information.

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
