# Peer review of "We need to talk about reliability: making better use of test-retest studies for study design and interpretation"

_PeerJ, doi:10.7717/peerj.6918_

## Round 0.1 · original submission · Major Revisions

This is an interesting manuscript which I enjoyed reading. Reviewer 1 is also positive about your manuscript. Reviewer 2 is still positive but has raised some areas for improvement which overlap with my comments below (such as R code/output for the package) and also raises the important question of normality. I will ask you to respond to all reviewer comments along with my comments below.

There are some issues with the terminology as I would use it (as a biostatistician) and I’ve pointed out some of these in the specific comments below. You also need to be careful to distinguish population parameters (normally shown using Greek letters) and sample statistics (normally shown using Roman letters).

I also wonder if it would be useful to include the commands from your R package, either in the manuscript or as a supplement, so that the reader can replicate your calculations and apply these to their own questions.

Line 26: The usual view of accuracy is related to measurement validity not reliability, although there are also less common views of accuracy as a mixture of validity and reliability. I’m not aware of any definitions of accuracy that would coincide with reliability alone. Also, reliability is not a measure of consistency, rather internal consistency, repeatability, and reproducibility can each inform a measure’s reliability. Note that under classical test theory, the approach you appear to be taking here, true scores cannot be observed, only measured, and so reliability can only be estimated and never known.

Line 28: If you mean the estimated reliability based on the sample, then this does approximate the population reliability; but if this is referring to the population’s reliability, this is approximated by the (appropriate) intra-class correlation coefficient (as you go on note on Lines 113–114). In Classical test theory, reliability is by definition the ratio of true score variance and observed variance (as you later note on Lines 97–98).

Line 33: Apologies for the pedantry, but the “could” here also applies for any instance where reliability < 1.

Lines 37–38: I think this statement has potentially become questionable in more recent times with confidence (and Bayesian credible) intervals becoming much more central to inference in clinical research. Perhaps note that NHST is used by much clinical research (the “performed using” seems to suggest that this is all that is done to me).

Line 39: “is THEN rejected”.

Lines 44–45: I think this might be overreaching a little. There has been a great deal of discussion around the association between reliability and statistical power in psychology, for example, and you might like to provide some references to this or other articles looking at reliability in the design of studies.

Lines 45–46: I think this would only apply if you were using standardised effect sizes. Am I missing something here? Do you mean “detected” rather than “expected”? Okay, on Lines 181–183 you note this for correlations, so perhaps you just need to clarify here that you are talking about that and not effect sizes more generally. Later on you mention also SMD, where again this would apply. I think it would be worth clarifying which effect sizes you have in mind here as this does not apply to all.

Lines 46–47: Note that power is only a function of reliability if one of the true score or error score variance is assumed to be constant (and the direction of the relationship depends on which variance changes). Statistical power depends on the magnitude of observed variances, reliability on the ratio of true and true plus error score variances and these can move in the same or opposite directions.

Lines 49–50. There are several definitions of “Type III errors” but power is not connected to Kimball’s, which I believe concerns statisticians not understanding the actual question being posed by researchers.

Lines 50–51: Again, I think this is either overreaching or slightly casual in its language. If the data is random noise, there is no signal, not merely a difficult to detect signal.

Line 97: It might be worth stating that you are working from a classical test perspective somewhere around here.

Line 99: Note that Cronbach’s alpha is a lower bound on (unqualified) reliability, not an estimate per se.

Line 107: This is entirely up to you, but I wonder if a quick recap of the different forms of ICCs would be useful for the reader at this stage.

Line 109: Do you mean “one-way random” here. This matches the formula provided, but would be my last choice for test-retest reliability.

Line 110: Asymptotically, I believe the two-way fixed model is the most conservative.

Line 113: Perhaps worth noting that the “usually 2” would apply to this domain. (It would be very unusual in other domains, such as measuring physical activity, to use only two measurement occasions.)

Equation 3: You could index j as 1..2 given your comment on Line 113 above and indicate that this is for the most likely case of 2 measurements here. This is just a thought that might help the reader to follow the equations. Alternatively, you could echo the “usually 2” on Line 122. Note also that sigma refers to a population parameter and so is estimated rather than known (as you note on Line 119) which makes the equality here inappropriate (equation 4 notes this correctly).

Equation 4: Again uses sigma for a sample statistic which should be “s”. Note this also on Line 127.

Line 134: It might be worth noting differences in nomenclature (minimum detectable difference is also common).

Equations 5 and 6: While these are true for the population, the sample estimates will again need to use “s” rather than sigma.

Lines 183–184: Again, for correlations as effect sizes but not more generally.

Line 185: Not necessarily, precision, model complexity, and testing a study protocol are also valid approaches to determining the required sample size. Perhaps qualify this as “sample size is often determined using power analysis” or specify the context such that the statement would always be true.

Lines 185–186: This is pretty much the definition I would use for “power analysis” (with “of interest” added after “effect size”) so I’m not sure about the “often”. Note that the other options for sample size would fall outside of “power analysis” as I know the term. I’d also add given certain parameters (including alpha, beta, variances).

Lines 186–189: I would have regarded this as standard practice in my experience. In any case, if you say “it has been suggested” you’ll need to indicate by who.

Line 192: I’m not sure you need “respectively” here.

Line 193: Or ethical.

Lines 195–196: I think you need to make it clear here that this applies to statistically significant findings and not all findings.

Line 217: I think it is important not to overlook just how difficult this calibration could be in practice.

Lines 222–223: Of course it could also go in the other direction (I appreciate that you do note this as an example, but on average I would anticipate errors proportional to the mean over the opposite).

Lines 224–225: Not necessarily if the base study is in one well-defined region and the proposed study in another well-defined region with different effect sizes and error variances, this would not require any caution about either study but rather about the borrowing of the SEM.

Lines 226–231: My advice to researchers who I was working with would probably include some scenario analyses, which is an approach that could be mentioned here if you wished.

Line 233 and Equation 8: Remember that we do not know sigma from the original study, only its estimate through s.

Equation 9: Note again that population parameters and sample statistics are different. Also middle component should be SEM I think.

Lines 246–247: Do you mean external validity (generalisability) here?

Lines 299–301: Poor reliability is not a reason for being cautious in interpreting statistically significant results. Poor reliability (for a valid measure) simply means more participants are needed to show evidence of an effect, and for correlations and standardised effect sizes, these will be underestimated as you have noted.

Line 326: I get n=24 for this but you may have used another approach (23.00936 from pwr.r.test(r=sqrt(0.3),sig.level=0.05, power=0.8)). Note that you also need to give the level of significance here.

Line 333: I mostly agree with you, but the 95% CI for r is still useful in itself and the study as a whole for meta-analyses.

Line 357: Again, need level of significance. I’m not a fan of one-sided tests as they only make sense if an effect in the other direction would be treated the same as no effect, so I think it is important to ask whether or not a surprising/paradoxical effect would be mention-worthy.

Line 373: As for Line 357.

Lines 391–393: Power and significance levels needed.

Lines 415–416: This is perhaps the problem with Cohen’s d, the interpretation is always context dependent. Standardised mean differences greater than 0.8 will arise with regularity in some disciplines while remaining at least “large” in others. In short, I don’t think you can reject an effect size as implausible based on Cohen’s d alone.

Line 427: And adjustments. I think you need to be careful not to underestimate the importance of these and the challenges involved in their determination. This is perhaps where scenarios for these might be worth discussing as well. At the moment, the text suggests, to me, that this is much easier than I believe it is likely to be in practice.

Figure 2: I’d add more decimal places to avoid the 100%s for U3 and CLES.

Line 436: This again emphasises the importance of reporting 95% (or other) CIs. Note that reliability will only call into question null findings or positive findings from an underpowered study, not all findings, and poor reliability will only attenuate some effect sizes.

Lines 478–479: Again, for some effect sizes, a large number of underpowered studies using an unreliable measure will still allow a meta-analysis to produce a sensible estimate of the true effect. I think these statements are just a little too sweeping although I do agree with them in general.

Reviewer 1 ·

Basic reporting

This is an interesting and mostly useful paper. The declared aim is to use test-retest data and insights into experimental design to build meaningful power analysis. There is nothing in the paper this reviewer disagrees with and I found the article very well structured and written.

Experimental design

This is a methodological paper that relies on previously published data.

Validity of the findings

As a researcher working in PET,I find the author's quantitative insights matching with experience and I can confirm that indeed endogenous release studies with PET are notoriously underpowered (except for dopamine) and a general waste of money. One note of warning is on the TSPO (PBR28) data. There is a lot of literature on the biological validity of Vt with these tracers; the true concentration of reference for the calculation of Vt is the free plasma concentration of the radiotracer in blood. Hence both plasma radioactivity and free plasma fractions are required. Now, free plasma fractions do not usually change much across subjects/patients and are generally ignored except for these tracers as 1) there is a lot of TSPO in blood cells and 2) TSPO radiotracers seem to have affinity for plasma reactive proteins. However since free plasma fractions for these tracers are very small, they cannot be measured reliably enough; hence the excess variability of VTs and the pragmatic choice of within brain normalizations

Additional comments

No further comments

·

Basic reporting

This is a solid article on a very important topic. My main concern is that the ideas, technical terms and equations are difficult to digest, mainly because of a lack of illustrations. I think the impact of the paper could be greatly increased by providing detailed illustrations, to help readers digest all the information in the methods section, and to give them an intuition for the relationships among variables/measurements. A first illustration could show the different sources of variability with the matching symbols used in the equations. Illustrations of actual data from different experimental designs/statistical analyses are also needed.

In the introduction, the standard concerns about false positives and true positives are contrasted with the more unusual one of reliability. In addition, measurement precision is considered an important factor by some researchers. You can see references about accuracy in parameter estimation here:
[[https://garstats.wordpress.com/2018/08/27/precision/]]
In particular:
Peters, G.-J.Y. & Crutzen, R. (2017) Knowing exactly how effective an intervention, treatment, or manipulation is and ensuring that a study replicates: accuracy in parameter estimation as a partial solution to the replication crisis. PsyArXiv. doi:10.31234/osf.io/cjsk2.

It would be worth articulating how these notions relate to each other in the introduction:
- power
- reliability
- accuracy in parameter estimation
You do mention all 3 aspects in later sections but the point should be clarified in the introduction.

Experimental design

I would suggest adding R code in the text, following each example (tutorial style) or at least as an appendix.

Validity of the findings

Illustrations are also needed to let readers assess the match between real data and the proposed methods. As far as I can see, all the tools implemented in the R package assume normality. Do we have to worry about this reliance on non-robust stats? For instance, skewness and outliers tend to inflate variance estimates and lower effect sizes. Is that a problem when assessing results from relatively small n studies, especially in the absence of scatterplots in the original articles? Are there robust versions of ICC? I’m not expecting a robust version of all the tools presented here, but at least the lack of statistical robustness should be acknowledged.

Additional comments

## other points
Figure 1: I do not understand the description in the right panel. How can there be a true sample ES? This sentence does not clarify the point and the “respectively” structure is hard to read: “The population (Popn.) effect size (ES) and sample ES are of the underlying distributions from which the data are sampled and of the obtained sample respectively.”

“In psychometrics, reliability is often assessed using internal consistency” - can you provide a reference?

## typos
“USD 10 000” -> 10,000
“little consideration can given to” -> can be given to
“as an estimate the reliability/relative/absolute” -> “as an estimate of the...”

---

## Round 0.2 · Minor Revisions

As the reviewer has noted, the manuscript is much improved and will make an important contribution to the field. Along with their, relatively minor, suggestions, I will make some minor comments myself below to accompany their suggestions and look forward to quickly accepting a revised version that addresses these.

Figure 1’s caption and Line 112: These are the first instances of “ICC” in the manuscript, but the full form is not presented until Line 119.

Lines 120–122: You could, if you agree, note that this particular ICC does not have a specific notation using Shrout and Fleiss. Otherwise, I suspect some readers will feel that this was “omitted”.

Equation 2: I think you’re missing parentheses around the final term in the denominator “MSC−MSE” (i.e. this should be “k/n (MSC-MSE)”).

Line 130: The colon here (“…previous studies: rather…”) should perhaps be a semi-colon as this is continuing the sentence rather than introducing a list or definition (“…previous studies; rather…”). Similar points would seem to apply for Lines 59, 62, 67, 82, 132, 149 (where it could be moved to introduce the list, i.e. “…it can be: i) scaled…”), 161, 178 (arguably), 213, 340, 359, 405, 409, 489, 501, 511, and 515 where I would personally omit some of these colons, change some to commas, and change others to semi-colons. I appreciate that there are stylistic elements here and I don’t think understandability is threatened in any case, so I’ll leave these suggestions entirely to your discretion.

Line 134: You could add a reference here that supports this approach for the interested reader (without going into any more detail than you already do).

Line 150: “…as an estimate OF the relative uncertainty…” (but depending on how you address the reviewer’s comment about Line 149)

Line 151: “…as an estimate OF the absolute…” (as above)

Line 217: I’d be inclined to delete “all” from “…such that all interesting effects…” as this could be (mis)read as an absolute (i.e. 100%) chance of reliably (whatever this means to the reader) detecting this effect. Alternatively, you could instead change “reliably detected” on Line 218 to “likely to be detected” to emphasise the risk of false negatives for what would be interesting effect sizes.

Lines 224–225: I think you could clarify here that the low-power => effect size inflation is specific to statistically significant findings only, e.g. “…avoid effect size inflation FOR STATISTICALLY SIGNIFICANT RESULTS due to low power…”

Lines 321–322: Would this be clearer as “…exhibits low measurement error, but EVEN lowER inter-individual heterogeneity, resulting in low reliability.” which draws attention to the relative variability (error and heterogeneity) that results in the ICC of 0.32 (Line 334)?

Lines 322–323: And similarly, “…shows high error variance, but EVEN highER inter-individual heterogeneity, resulting in high reliability.” (approx. 0.90 from Line 431)

Figure 4: While I appreciate that the rounded values are “100%” in figure 5 for the rightmost panel, you could consider using “>99%” instead to make it clearer that these are not literally 100%.

Line 479: I wonder if “… , and should usually, …” here would be easier to read as “… , as is appropriate, …”.

Line 524: I think many readers would be helped by showing the full form of WEIRD here (“Western, Educated, Industrialized, Rich, and Democratic”) as you do for QRPs on Line 527.

·

Basic reporting

Fine, only some typos and potential writing issues:

49: there there has been
107: “True scores can never be directly observed, but only observed with different degrees of error.” -> but only ESTIMATED
149: “scaled to the variance of the sample as an estimate the reliability” and subsequent points -> as an estimate OF the reliability / as a reliability estimate / TO estimate reliability...
203: “when performing power analysis” -> a power analysis / power analyses
375: “larger than what what was considered”
376: “insignificant” -> non-significant
491: “reaching the the data”
566: “an equal equal radioligand“

Experimental design

Excellent.

Validity of the findings

Excellent.

Additional comments

The new version is much clearer. Your article presents a tremendous amount of work and I hope it will contribute to researchers spending more time thinking about statistical issues before they run their studies. The revised discussion provides a rich description of the multiple angles to consider when designing an experiment, with plenty of important references for curious readers - great job!

Thank you for adding the figures and the code.
I only have a few minor comments, the main one about the caption of figure 2.

Figure 2 is very nice, but the caption needs to be improved. For instance, this sentence is very difficult to understand: “Low reliability within groups is increased for the total sample in a comparison, given a sufficiently large effect size, due to the larger variance of the combined sample.” One suggestion would be to refer to the 2 panels Group 1 and Group 2, then mention their combination, pointing to the third panel (needs a name), and explain what happens to the ICC. For the right panels, I still find the True/Measured dichotomy hard to understand in relation to the Popn/sample one. Instead of True/Measured, would the distinction be better described as without/with measurement error? Also, a general title is missing.

---

## Round 0.3 · accepted · Accept

Thank you for your revisions, which I think address all of my and the reviewer’s comments. Thank you also for your positive words about and constructive responses to those comments during the review process. I am glad, as I am sure the reviewers are also, that you’ve found the review process to be helpful. I am confident that your manuscript will lead to much thought and discussion amongst researchers, and, in time, this will be reflected in citations showing the impact of your work. Well done!

I have made a few very small comments below for you to consider when looking at the proof of your article, none of which are worth holding up the acceptance of your manuscript for.

1) The manuscript refers to “Cohen’s D” in several places, and while I definitely wouldn’t call this incorrect, I realise that I cannot recall seeing the “d” for this as a capital letter before. While I wouldn’t use Wikipedia as an authority, you can see this on https://en.wikipedia.org/wiki/Effect_size#Cohen%27s_d. Similarly, given your obvious interest in typography, you could make the 3 in Cohen’s U3 a subscript (see, for example, https://rpsychologist.com/d3/cohend/).

2) Apologies for not commenting on this before, but should the sentence on Page 4: “Similarly, even if a measure is extremely accurate, it is still incapable of meaningfully distinguishing between individuals who all obtain the same score.” say “possess almost identical scores” rather than “obtain the same score”. I interpret this as referring to their true scores (not their measured scores, although these points will be almost identical with extreme accuracy) and the “meaningfully” earlier in the sentence requires these to be non-identical (otherwise the lack of capability to distinguish is absolute irrespective of accuracy). I don’t think what is currently written in the manuscript is incorrect, but these (very pedantic) points occurred to me on re-reading.

3) Also on re-reading, I think equation (3) should be dividing by n(k-1) and not n. In Baumgartner, this doesn’t matter as k=2 there and so dividing by n is equivalent to dividing by n(k-1), i.e. n(2-1) which simplifies to n, but you are giving the more general case here. See equation 18 in https://tutcris.tut.fi/portal/files/999666/wallius.pdf for example.

4) On page 5, in the last sentence of the penultimate paragraph, you say: “…since even with a reliability of 0.9, the standard error is almost a third the size of the standard deviation.”, which could potentially confuse some readers as this is the “standard error OF MEASUREMENT” (which is not the one that necessarily comes to mind when you read “standard error” without qualification). You could, if you agree and want to, add “of measurement” here to clarify this point.

5) Page 6, you have a spurious space before the semi-colon in “(Spearman 1904 ; Nunnally 1970).

6) Page 8 has a heading including “(t-tests)” (with a hyphen) but the paragraph below has text including “t tests” (no hyphen). Either is fine to me, but you might want these to be consistent.

7) Figure 2. The three instances of “ICC=0.xx” seem a little blocky to me in the generated PDF, although they look perfectly fine in the original JPEG. This could also be a result from some configuration on my computer, although I can’t think of what that would be.

8) Pages 10, 12, and 16. While “COV” is certainly used for “coefficient of variation”, “CV” is much more common. (Google searches of “coefficient of variation” with each produced ~300k hits for the former and ~8m hits for the latter.)

9) The right-most part of Figure 3 was cut off in the generated PDF for me (looks like the scaling was off here).

10) On Page 14, you say “…Consideration of the reliability is another important, and related, factor.” where it seems either the “the” is unneeded or words are missing that would give a context to “reliability”.

#